# Therapeutic Options for Crigler–Najjar Syndrome: A Scoping Review

**DOI:** 10.3390/ijms252011006

**Published:** 2024-10-13

**Authors:** Vanessa Sambati, Serena Laudisio, Matteo Motta, Susanna Esposito

**Affiliations:** Pediatric Clinic, Department of Medicine and Surgery, University Hospital of Parma, 43126 Parma, Italy; vanessa.sambati@unipr.it (V.S.); serenarosa.laudisio@unipr.it (S.L.);

**Keywords:** Crigler–Najjar syndrome, *UGT1A1* mutation, liver transplantation, gene therapy, hyperbilirubinemia

## Abstract

Crigler–Najjar Syndrome (CNS) is a rare genetic disorder caused by mutations in the *UGT1A1* gene, leading to impaired bilirubin conjugation and severe unconjugated hyperbilirubinemia. CNS presents in the following forms: CNS type 1 (CNS1), the more severe form with the complete absence of *UGT1A1* activity, and CNS type 2 (CNS2), with partial enzyme activity. This narrative review aims to provide a detailed overview of CNS, highlighting its clinical significance and the need for new, more effective treatments. By summarizing current knowledge and discussing future treatments, this article seeks to encourage further research and advancements that can improve outcomes for CNS patients. The literature analysis showed that CNS1 requires aggressive management, including phototherapy and plasmapheresis, but liver transplantation (LT) remains the only definitive cure. The timing of LT is critical, as it must be performed before the onset of irreversible brain damage (kernicterus), making early intervention essential. However, LT poses risks such as graft rejection and lifelong immunosuppression. CNS2 is milder, with patients responding well to phenobarbital and having a lower risk of kernicterus. Recent advancements in gene therapy and autologous hepatocyte transplantation offer promising alternatives to LT. Gene therapy using adeno-associated virus (AAV) vectors has shown potential in preclinical studies, though challenges remain in pediatric applications due to liver growth and pre-existing immunity. Autologous hepatocyte transplantation avoids the risk of rejection but requires further research. These emerging therapies provide hope for more effective and less invasive treatment options, aiming to improve the quality of life for CNS patients and reduce reliance on lifelong interventions.

## 1. Introduction

Crigler–Najjar syndrome (CNS) is a rare autosomal recessive disorder characterized by severe unconjugated hyperbilirubinemia. CNS results in nonhemolytic jaundice, with its most serious complication being bilirubin-induced neurologic dysfunction. This occurs when bilirubin crosses the blood–brain barrier and binds to specific brain tissues. CNS is associated with various mutations in the bilirubin uridine diphosphoglucuronate glucuronosyltransferase coding gene (bilirubin-UGT, also known as *UGT1A1*) [1].

Two forms of CNS are recognized based on the extent of *UGT1A* activity loss. Type I disease (CNS1) is characterized by severe jaundice and a high risk of neurologic sequelae (kernicterus), resulting from the complete absence of enzymatic activity; type II disease (CNS2) is a milder form with decreased enzyme activity, lower serum bilirubin levels, and little to no risk of kernicterus [2].

Current therapeutic options for CNS include phototherapy, phenobarbital administration, and liver transplantation. Additional approaches include liver cell transplantation and gene therapy. This narrative review aims to provide a detailed overview of CNS, highlighting its clinical significance and the need for new, more effective treatments. By summarizing current knowledge and discussing future treatments, this article seeks to encourage further research and advancements that can improve outcomes for CNS patients.

## 2. Methodology

Two of the authors (SL and MM) reviewed the literature by searching the Medline database via the PubMed interface (https://pubmed.ncbi.nlm.nih.gov/ on accessed 12 September 2024) and Google Scholar (https://scholar.google.com/ accessed on 12 September 2024), selecting articles published between January 2000 and January 2024. Case reports and clinical trials on standardized and experimental therapies were also collected to highlight currently available treatment options. The search utilized controlled vocabulary and key terms (MESH indexed: “Crigler–Najjar syndrome” and/or “Crigler Najjar syndrome”). Articles were selected based on the following predefined inclusion criteria: (a) patient population (pediatric patients diagnosed with CNS); (b) the primary focus of the article (pathogenesis, therapy, outcome, future perspectives); (c) availability of relevant data within the article; (d) publication period (2000–2024); (e) human subjects; and (f) language (English). The exclusion criteria were (a) non-English language; (b) lack of a full abstract; (c) unsupported data; and (d) articles reporting novel mutations without a description of clinical or therapeutic implications. After the initial searches, 55 articles were chosen based on title and abstract screening. Of these, 30 studies specifically focused on CNS therapy were selected for their relevant content, which reported on CNS treatments and outcomes. Narrative reviews, studies reporting genetic mutations without phenotypic impact, and reports that combined data from both adult and pediatric patients without distinction were excluded. Additional sources cited within the selected articles were also reviewed, resulting in a total of 64 articles referenced in this systematic review. The cited articles are organized by title, location, study design, population, study objectives, summary of results, and conclusions.

## 3. Insights into Crigler–Najjar Syndrome (CNS)

### 3.1. Epidemiology

CNS is an extremely rare disorder, affecting approximately 0.6 to 1 in every 1 million newborns worldwide and fewer than 1 in 100,000 individuals in Europe [3]. Both sexes are equally affected, and the condition occurs across all ethnic groups. However, it is more prevalent in genetically segregated populations, such as the Old Order Amish and Mennonite communities, as well as among children born to consanguineous parents [4].

### 3.2. Genetics

CNS types I and II are caused by mutations in the *UGT1A1* gene, located on chromosome2 q37, which result in impaired enzymatic activity. UGT proteins are divided into two families, UGT1 and UGT2, based on amino acid sequence similarity. The *UGT1* family includes nine functional enzymes (*UGT1A1* to *UGT1A10*) and four pseudogenes, all encoded by a single gene on chromosome2 q37. Each *UGT1A* isoform has a unique first exon, followed by exons 2 to 5 [5].

CNS is related to Gilbert syndrome (GS): they are both genetic disorders caused by mutations in the *UGT1A1* gene, which affects bilirubin metabolism. In CNS, there is a severe deficiency or complete absence of the enzyme UGT1A1, leading to very high levels of unconjugated bilirubin and, in severe cases, life-threatening complications such as kernicterus. In contrast, GS involves a mild deficiency of the same enzyme, resulting in slightly elevated bilirubin levels and occasional jaundice, but typically without serious health issues. Both conditions share a common genetic basis but differ in the severity of enzyme deficiency and clinical outcomes [6].

*UGT1A1* is predominantly expressed in the liver, and genetic testing for *UGT1A1* mutations is used to confirm diagnoses of GS, CNS2, or CNS1 in patients with persistent unconjugated hyperbilirubinemia without hemolysis. Identifying these patients is crucial for appropriate management, early treatment initiation, and for offering genetic counseling to family members [7].

The genotype–phenotype correlation is complex, as variations in enzyme activity may also result from genomic, transcriptional, translational, or post-translational changes. In CNS1, there is a complete loss of *UGT1A1* activity due to deletions, insertions, missense mutations, or premature stop codons, resulting in severe unconjugated hyperbilirubinemia [8].

Most CNS2 patients have homozygous or compound heterozygous missense mutations, reducing enzyme activity to less than 10% of normal. This residual activity can often be enhanced with phenobarbital treatment [9]. Genetic studies have shown that chain-terminating mutations are more common in CNS1, while missense mutations are more frequent in CNS2 patients [7,10].

To investigate *UGT1A1* mutations in GS and CNS2, Sun et al. studied 95 adult Chinese patients with hereditary unconjugated hyperbilirubinemia. Their analysis revealed 192 mutations across six sites. The most common mutation in GS was c.-3279T>G in the PBREM region (36.3%), followed by A(TA)7TAA (30.6%). In CNS2 patients, p.G71R was the most frequent mutation (38.2%), followed by c.-3279T>G and p.Y486D [10]. Similarly, Li et al. sequenced *UGT1A1* in 11 unrelated Chinese CNS2 patients and found two cases with only a single heterozygous variant [11].

Given these findings, it is more accurate to refer to *UGT1A1*-associated diseases as a spectrum of conditions rather than distinct diagnoses, acknowledging variable penetrance and expressivity. Many cases with bilirubin levels and clinical features that fall between GS and CNS2 or between CNS-II and CNS-I have been reported [5,7,10,12].

### 3.3. Pathophysiology

Bilirubin is a product of heme degradation, with 80–90% derived from hemoglobin in aging red blood cells. In the reticuloendothelial system, heme is broken down by heme oxygenase, producing carbon monoxide and biliverdin. Biliverdin is subsequently converted to bilirubin by biliverdin reductase. Unconjugated bilirubin, which is not water-soluble, binds to albumin for transport to the liver, where it undergoes conjugation. Once conjugated, bilirubin is excreted into bile and enters the intestines. In the gut, bacteria convert conjugated bilirubin into urobilinogen. Some urobilinogen is reabsorbed into the blood via the enterohepatic circulation and excreted in the urine as urobilin (its oxidized form), giving urine its yellow color. Additionally, urobilinogen is further converted by colonic bacteria into stercobilinogen, which is oxidized to stercobilin, the compound responsible for the brown color of feces. This process highlights the importance of bilirubin’s transport by albumin in its unconjugated form and the role of gut bacteria in its final excretion forms. CNS results from impaired bilirubin conjugation. In CNS1, unconjugated bilirubin levels typically range from 20 to 25 mg/dL but can reach as high as 50 mg/dL. Without *UGT1A1* activity, bilirubin accumulates at a constant rate of 3.8 ± 0.6 mg/kg/day [13]. In CNS2, also known as Arias’ disease [14], *UGT1A1* activity is reduced but not completely absent, leading to less severe hyperbilirubinemia, with levels below 20 mg/dL, though they can rise to 40 mg/dL during acute exacerbations [15].

### 3.4. Clinical Features

CNS1 presents with severe unconjugated hyperbilirubinemia beginning in the neonatal period and persisting throughout life. The primary complication is bilirubin-induced neurological dysfunction (BIND), particularly in the neonatal period [4]. Patients with CNS1 do not respond to phenobarbital and require long-term phototherapy to keep serum bilirubin levels within safe limits. CNS2 presents with varying phenotypes, and some patients maintain safe bilirubin levels without chronic treatment, while others may need phenobarbital or additional interventions during acute exacerbations triggered by infection, hemolysis, or biliary obstruction.

Jaundice in CNS usually appears when bilirubin levels exceed 5 mg/dL and is most easily visible in the fingerprint area, progressing in a cephalon–caudal pattern. Hyperbilirubinemia in CNS1 typically ranges from 10 to 25 mg/dL during the first 10 days of life, with an increased risk of neurological damage as bilirubin levels rise. Both total serum bilirubin (TSB) and the bilirubin/albumin ratio (B/A) are used to predict neurotoxicity. TSB ≥ 30 mg/dL and a B/A ratio ≥ 1.0 mol/mol are considered absolute thresholds for neurotoxicity [4,16].

If untreated, CNS can lead to acute bilirubin encephalopathy, kernicterus, or chronic bilirubin-induced neurological dysfunction (BIND), which manifests with symptoms like altered consciousness, abnormal tone, and impaired auditory responses [17]. Chronic bilirubin encephalopathy, or kernicterus, is characterized by choreoathetoid cerebral palsy, sensorineural hearing loss, palsy of vertical gaze, and dental enamel hypoplasia. BIND can occur in adolescence or adulthood, and a broader term, kernicterus spectrum disorder (KSD), has been proposed to unify diagnoses based on clinical and pathophysiologic criteria [18].

Historically, CNS was considered a condition with jaundice but without liver damage. However, recent reports suggest that hepatic fibrosis is present in 40–60% of patients undergoing liver transplantation, with fibrosis severity correlating with bilirubin concentration and age [19].

### 3.5. Quality of Life

The long-term impact of CNS on quality of life is not fully understood, but the burden of phototherapy is substantial for both patients and caregivers. For CNS1 patients, phototherapy requires whole-body exposure for 10–12 h per day starting from birth [20]. Although it does not appear to interfere with circadian rhythms [21], it significantly restricts travel and social activities, imposing a heavy burden on patients and caregivers [22]. While liver transplantation is a definitive treatment, it comes with risks such as donor availability, potential graft failure, and the lifelong need for immunosuppressive drugs.

### 3.6. Pregnancy and Fetal Risks

Pregnancy in women with CNS is rare and presents unique challenges. The risk of fetal kernicterus and rising maternal bilirubin levels during pregnancy are major concerns. Unconjugated bilirubin crosses the placenta by passive diffusion, and in cases of elevated maternal bilirubin, the fetus is at risk for neurological complications and kernicterus [22]. Regular monitoring and adjusting phototherapy duration are recommended to maintain maternal bilirubin levels below 11.7 mg/dL, with a B/A ratio below 0.5 [23]. In CNS2 patients, combining phototherapy and phenobarbital can keep bilirubin levels within safe ranges [24]. Sequential treatment with phototherapy during the first trimester and phenobarbital in subsequent trimesters has been successfully used to manage bilirubin levels in pregnant women with CNS2 [25]. Hannam et al. reported two infants born to women with CNS1. Both required exchange transfusions after birth, and one developed sensorineural hearing loss at seven months, despite phototherapy and albumin infusions during pregnancy [26].

## 4. Diagnosis of Crigler–Najjar Syndrome (CNS)

CNS should be suspected in any newborn with persistent jaundice and severe unconjugated hyperbilirubinemia, particularly after excluding common causes such as physiological jaundice, breast milk jaundice, polycythemia, systemic illness, and other inherited disorders of bilirubin metabolism [1]. Early recognition is crucial, as untreated CNS can lead to serious complications like kernicterus and long-term neurological damage.

The definitive diagnosis of CNS is made through molecular genetic testing for mutations in the *UGT1A1* gene, which encodes the enzyme responsible for bilirubin conjugation. CNS can result from a variety of genetic abnormalities, including missense and nonsense mutations, insertions, deletions, and splicing abnormalities that affect any of the five exons of the *UGT1A1* coding region. Therefore, it is important to sequence not only all exons but also the flanking intronic regions using Sanger sequencing or next-generation sequencing techniques. Additionally, sequencing the promoter region, particularly the TATAA box, is essential because the presence of a *UGT1A1* polymorphism such as *UGT1A128* (associated with GS) can further reduce the expression of the enzyme, exacerbating hyperbilirubinemia [1].

A key clinical feature distinguishing CNS1 from CNS2 is the serum bilirubin concentration. CNS2 typically presents with lower bilirubin levels than CNS1, although during acute exacerbations, the values can overlap. To aid in differentiating CNS2 from CNS1, a trial of phenobarbital therapy can be employed. In CNS2, phenobarbital typically lowers serum bilirubin levels by about 25%, while CNS1 patients show no significant response [1]. However, confirmation through genetic testing is required for a definitive diagnosis. CNS1 is associated with more severe mutations, including premature stop codons, frameshift mutations, or non-synonymous (missense) mutations that lead to the substitution of a single amino acid, resulting in the complete loss of *UGT1A1* enzyme activity [1,27]. On the other hand, CNS2 is typically caused by missense mutations that reduce the enzyme’s catalytic activity but do not abolish it entirely. If a patient with CNS2 also carries a Gilbert-type *UGT1A128* promoter mutation, hyperbilirubinemia may be further aggravated because of the decreased expression of the enzyme [1]. Historically, additional diagnostic methods included high-performance liquid chromatography (HPLC) or thin-layer chromatography of bile samples obtained from the duodenum. These tests were used to assess bilirubin conjugation patterns, although their role has diminished with the advent of advanced genetic testing [28].

In summary, a stepwise diagnostic approach for CNS includes the clinical assessment of jaundice, exclusion of other causes of hyperbilirubinemia, the trial of phenobarbital in suspected CNS2 cases, and confirmation through comprehensive molecular genetic testing of the *UGT1A1* gene. Early diagnosis and differentiation between CNS types are essential for guiding appropriate treatment and long-term management strategies.

## 5. Therapeutic Approach to Crigler–Najjar Syndrome (CNS)

Table 1 and Table 2 summarize the therapeutic strategies for CNS and the main published studies on treatment [29,30,31,32,33,34,35,36,37,38,39]. While phototherapy and phenobarbital remain the primary treatments for CNS, especially in CNS1 and CNS2, ongoing research into alternative therapies offers hope for improved management of the condition. Genetic analysis plays a critical role in guiding treatment decisions and tailoring therapies to individual patients, as CNS phenotypes can vary widely. However, liver transplantation remains the only definitive cure for CNS1, and early identification and management are essential to prevent complications like kernicterus.

### 5.1. Phototherapy

Phototherapy is the cornerstone of CNS treatment, particularly during infancy and childhood, and its introduction has significantly altered the disease’s trajectory [29]. Phototherapy works by converting bilirubin IX-alpha-ZZ into its configurational isomers (such as lumirubin), which can then be excreted in bile without the need for conjugation [30]. Treatment should begin as soon as possible after birth, ideally following an exchange transfusion if required, and should be administered for an average of 12.4 ± 0.8 h per day, including during the night.

To improve the effectiveness of night therapy, ursodeoxycholic acid (15–30 mg/kg/day) and a lipid-rich bedtime snack are recommended to stimulate bile flow and hepatic clearance of lumirubin. While various phototherapy systems provide high irradiance over large body surfaces, the therapy becomes less effective at puberty because of factors like skin thickening, increased pigmentation, and a reduced surface area-to-body mass ratio. During adolescence, bilirubin levels can reach dangerous levels, necessitating alternative or adjunctive therapies [4].

Patients with CNS2 have milder jaundice and can typically be treated with shorter phototherapy sessions, improving their quality of life. In contrast, CNS1 patients require immediate and sustained phototherapy to reduce the risk of kernicterus.

### 5.2. Phenobarbital

Phenobarbital is the first-line therapy for CNS2 patients, typically used to prevent acute increases in bilirubin levels during illness or stress. The drug induces *UDPGT* activity, enhancing bilirubin clearance by increasing hepatic uptake, storage, and excretion [31].

Chronic treatment is recommended for patients with serum bilirubin levels exceeding 15 mg/dL or for those experiencing significant jaundice that affects their quality of life. Standard doses are 2 mg/kg, two–three times per day in children, or 60–180 mg/day in adults, divided into two doses [7]. However, bilirubin levels may fluctuate, and responses to phenobarbital can vary among patients.

Some patients do not respond to phenobarbital. For example, Shi et al. [32] described a Bangladeshi patient with CNS2 who did not respond to phenobarbital (30 mg/day) despite having bilirubin levels typical of CNS2. Genetic analysis revealed heterozygous mutations, including an insertion in the HNF-1α site and a deletion in exon 1 of the *UGT1A1* gene. The mutation in the HNF-1α binding site rendered the allele unresponsive to *UGT1A1*-inducing drugs, including phenobarbital. In such cases, confirming the presence of a mutated HNF-1α site should discourage the use of phenobarbital [32].

In contrast, Liu et al. [7] reported an unusual case of CNS in a Chinese boy with severe jaundice and kernicterus. Despite initial high bilirubin levels (40.4 mg/dL), the boy responded dramatically to phototherapy and phenobarbital, with serum bilirubin dropping to 5.4 mg/dL within days. Although his bilirubin levels suggested CNS1, his responsiveness to phenobarbital indicated CNS2, highlighting the importance of genetic analysis in confirming the diagnosis.

Gailite et al. [33] reported another interesting case of a 17-year-old male initially diagnosed with Gilbert Syndrome due to a homozygous *UGT1A1* promoter polymorphism (A(TA)7TAA). However, after developing jaundice during puberty and responding to phenobarbital, the patient was reclassified as CNS2, despite having bilirubin levels consistent with CNS1. Genetic testing revealed four *UGT1A1* variants, underscoring the importance of genotype–phenotype correlation for accurate classification and treatment decisions [33].

These cases highlight the variability in CNS phenotypes and the necessity of genetic testing for proper diagnosis and treatment planning.

### 5.3. Alternative Treatments

While there is no currently authorized medicinal product for the treatment of CNS, several alternative therapies have been investigated over the years.

Ihara et al. [34] followed a Japanese CNS1 patient from birth to age 19. After phototherapy became less effective at age 16, the patient was treated with plasmapheresis combined with an anion-exchange resin. This method removed approximately 98% of the daily bilirubin production, temporarily reducing bilirubin levels. However, its efficacy was inversely proportional to plasma volume, and the patient ultimately required a liver transplant at age 17 [34].

Schwegler et al. [35] reported a 17-year-old girl with CNS2 who showed a significant reduction in serum bilirubin levels after treatment with clofibrate. Building on this, Yilmaz et al. [36] conducted a trial in two CNS2 patients using fenofibrate (250 mg/day for one month). However, fenofibrate did not significantly reduce serum bilirubin levels (14.2 mg/dL pre-treatment vs. 14.5 mg/dL post-treatment).

Hafkamp et al. [37] conducted a randomized, placebo-controlled, double-blind, cross-over trial in 16 CNS patients to evaluate the effect of orlistat, a fat absorption inhibitor. Orlistat significantly reduced plasma unconjugated bilirubin concentrations (by approximately 43%) in 40% of the patients, particularly in those with lower dietary fat intake and BMI. However, the clinical response was not correlated with age, sex, CNS type, or co-treatments (phototherapy or phenobarbital).

A Cochrane review [38] examined the therapeutic potential of metalloporphyrins, which are heme analogs that inhibit heme oxygenase, the enzyme responsible for converting heme to bilirubin. Although some studies suggested that metalloporphyrins could lower plasma bilirubin levels and reduce the need for phototherapy, their use is limited because of adverse effects like photosensitivity and anemia. The review concluded that there is insufficient evidence to recommend metalloporphyrins as a routine treatment for neonatal unconjugated hyperbilirubinemia.

## 6. Liver Transplantation

Liver transplantation (LT) is currently the only definitive treatment for CNS1. When performed early, LT has shown excellent outcomes, significantly improving quality of life and preventing the severe complications associated with the disease, such as kernicterus, with relatively few complications [20,40,41].

The three principal types of liver transplantation used to treat CNS1 patients are (1) Orthotopic Liver Transplantation (OLT), the most common approach, which involves replacing the entire native liver with a whole or partial graft from either a living or deceased donor; (2) Auxiliary Liver Transplantation (ALT), where a segment of the donor’s liver is transplanted while retaining the patient’s native liver; and (3) Auxiliary Partial Orthotopic Liver Transplantation (APOLT), which is similar to ALT, but only a partial liver graft is transplanted alongside the patient’s own liver. There is no difference between ALT and APOLT in surgical strategies, and the graft can come from a deceased donor (whole or split) or a living donor, typically a related donor such as a parent [42].

In CNS1, LT is usually recommended after the first year of life, particularly in adolescence, because its results improve. Recipients below 1 year fare worse in general. Moreover, as children grow, the increased body mass, thickening skin, and pigmentation decrease phototherapy efficiency, leading to longer hours under phototherapy lights, often during both day and night, which significantly impacts the patient’s quality of life [42].

While LT offers a cure for CNS1 by providing a liver that can conjugate bilirubin, it is a highly invasive procedure with several limitations. Unfortunately, the number of pediatric donors limits the availability of liver transplantation. However, surgical strategies like liver splitting can offer great advantages to pediatric recipients. Moreover, the risk of graft rejection and the need for life-long immunosuppression, which comes with associated risks like infections and malignancies, represent possible obstacles to obtaining favorable results. Thus, determining the optimal timing for LT is crucial—delaying it too long may result in irreversible neurological damage, while performing it too early increases exposure to the risks of surgery and immunosuppression.

Akdur et al. [43] reported five pediatric CNS1 patients who underwent LT. They emphasized that OLT is curative for CNS1, with four of the five patients showing positive outcomes without neurological deficits. However, one patient, a 2-month-old infant, died post-OLT because of complications related to kernicterus. This infant, the youngest CNS1 patient to undergo LT, had undergone plasmapheresis because of extremely high bilirubin levels (30 mg/dL) and suspected bilirubin encephalopathy. Kernicterus developed post-transplant, leading to death, underscoring that once full-blown kernicterus develops, its effects may be irreversible even after transplantation [43,44]. This highlights the importance of initiating medical treatments such as plasmapheresis and phototherapy at the first signs of bilirubin encephalopathy and considering early LT to prevent further neurological damage.

Other reports also support early intervention. Tu et al. [45] described an 18-month-old child with CNS1 who underwent LT immediately after developing symptoms of kernicterus. The child’s total bilirubin level increased to 742.0 µmol/L before the operation, but post-transplant, bilirubin levels normalized, and mental function recovered without further signs of neurological damage. Schauer et al. [46] described three CNS1 patients aged 7, 12, and 4 years who underwent OLT before irreversible brain damage occurred. The 7-year-old exhibited early signs of brain injury (retardation) and had poor compliance with phototherapy. After LT, mental and physical development improved significantly. The 12-year-old had more advanced hyperbilirubinemia-related brain damage, with apathy, slurred speech, and impaired coordination, but these symptoms improved post-transplant. Although not fully recovered, the patient was able to attend school in the appropriate grade level. The 4-year-old, who showed no signs of neurological injury, had an excellent post-operative outcome, further underscoring the importance of early LT before severe neurological damage occurs.

In regions where cadaveric donor organs are scarce, living-related liver transplantation (LRLT) has become a viable alternative. Shurafa et al. [47] reported a case series of six CNS1 children from Saudi Arabia, three of whom underwent LRLT. In countries where CNS1 incidence is higher because of consanguineous marriages, LRLT is a critical option, providing a timely solution for patients awaiting transplantation. The outcomes in these cases were successful, demonstrating that LRLT can be an effective and reliable treatment option in countries with donor organ shortages.

Innovations such as split liver transplantation and domino transplantation are also being explored to increase the availability of donor organs. In 2002, Schauer et al. [48] reported the first case of simultaneous split liver transplantation in two brothers (aged 4 and 12) with CNS1, who received liver segments from the same donor. The younger brother, with no neurological deficits before surgery, had a completely uneventful recovery. The older brother, who had moderate brain damage (apathy, slurred speech, and reduced mental activity), showed significant improvement post-transplant, with enhanced mental and physical development as bilirubin levels normalized. Domino liver transplantation has been proposed as another option, particularly for certain non-cirrhotic metabolic liver diseases (NCMLD). Combining the APOLT technique with domino liver transplantation, where a CNS1 patient’s liver could be used as a donor for patients with NCMLD, offers an innovative strategy for treatment. Govil et al. described the first domino–auxiliary transplant case for a child with CNS1 who received a liver from a child with propionic acidemia, demonstrating the feasibility of this approach [49]. Dong et al. [50] also reported successful domino–APOLT in three CNS1 patients and described how the metabolic defect in the recipient’s liver was fully compensated by the donor liver.

While LT can successfully cure CNS1 by providing functional UGT1A1 enzyme activity, it is not without risks. Complications such as graft failure, portal vein thrombosis, and biliary stenosis have been reported. In one domino-APOLT case, a CNS1 patient developed portal vein thrombosis 2 days post-transplant, requiring reoperation, and later developed biliary stenosis that was successfully treated with percutaneous balloon dilatation [50]. Despite these risks, the metabolic defect in all cases was successfully corrected.

Overall, LT remains the only definitive cure for CNS1, offering a solution for patients who can no longer be managed with phototherapy or phenobarbital. The choice between orthotopic, auxiliary, or living-related liver transplantation depends on the availability of donor organs and the individual circumstances of the patient. Although LT carries risks such as graft rejection, life-long immunosuppression, and surgical complications, it has been shown to effectively prevent irreversible neurological damage and restore normal bilirubin metabolism, greatly improving patients’ quality of life. Further advancements in transplant techniques, including domino and split liver transplants, offer hope for expanding donor organ availability and improving outcomes for CNS1 patients.

## 7. Liver Cell Transplantation

Liver cell transplantation (LCT) is a novel and less invasive therapeutic approach for treating CNS, particularly CNS1. LCT was first experimented with in Gunn rats, where isolated normal hepatocytes were introduced into the portal venous system, allowing these cells to integrate into the liver and become part of the hepatic cords [51]. Studies demonstrated that normalization of bilirubin levels in the Gunn rat model could be achieved with only 12% of the total liver mass expressing functional *UGT1A1* [52]. This suggests that small amounts of transplanted liver tissue in humans can alleviate the underlying metabolic deficiency.

In CNS1 patients, LCT involves the transplantation of donor hepatocytes expressing functional *UGT1A1* into the liver, with immunosuppressive therapy to prevent rejection. The less invasive nature of LCT offers advantages over whole-liver transplantation (LT), but the procedure has challenges, including the limited availability of cryopreserved hepatocytes and technical difficulties in ensuring adequate cell engraftment. Additionally, the transplanted cells typically function for only 6 to 9 months, making the treatment a temporary solution rather than a permanent cure [53]. Ongoing research is needed to optimize cell survival and long-term engraftment to improve the efficacy of LCT.

The first human LCT was performed in 1997 on a 5-year-old boy with ornithine transcarbamylase deficiency. In this case, splenic transplantation of adult hepatocytes helped control hyperammonemia and correct the metabolic defect temporarily, serving as a bridge to liver transplantation [54]. The following year, Fox et al. [55] treated a 10-year-old girl with CNS1 using LCT, which resulted in lowered bilirubin levels and reduced phototherapy needs. This was attributed to increased *UGT1A1* activity, reaching up to 5.5% of normal enzyme function.

In a case series by Quaglia et al. [56], seven children with metabolic disorders underwent LCT, including a 1-year-old CNS1 patient who received 4.34 billion hepatocytes through the inferior mesenteric vein. This resulted in a 50% reduction in bilirubin levels and decreased phototherapy requirements. However, the effect diminished after 7 months, and the patient eventually underwent liver transplantation. Similar reports in the literature have shown that LCT can improve the quality of life in CNS1 patients by lowering bilirubin levels and reducing the time required for phototherapy [56].

Dhawan et al. [57] described two CNS1 patients (18 months old and 3 years old) who experienced a 50% and 30% reduction in serum bilirubin levels, respectively, following LCT. These cases further emphasize LCT’s role in temporarily managing hyperbilirubinemia and delaying the need for liver transplantation.

More recently, Ribes-Koninckx et al. [58] presented data from four pediatric patients with different metabolic disorders who received hepatocyte transplantation with cryopreserved human hepatocytes. Their study demonstrated the safety and efficacy of LCT as a bridge to LT, with one 7-month-old CNS1 patient showing a significant improvement in both neurological status and bilirubin levels. Despite severe jaundice and neurological symptoms (spasticity and axial hypotonia) due to kernicterus, LCT stabilized bilirubin levels at less than 50% of the initial values and reduced phototherapy from 24 to 12 h per day. Furthermore, neuroradiological improvements were noted on magnetic resonance imaging, and the patient’s clinical status significantly improved, showing the potential for LCT to enhance neurological outcomes.

In another recent study, Jorns et al. [59] reported on two CNS1 patients who underwent liver resection followed by LCT. The initial liver resection served as a regenerative stimulus, leading to a transient increase in hepatocyte growth factor, which facilitated the engraftment of transplanted hepatocytes. This dual approach, combining partial hepatectomy and LCT, reduced serum bilirubin by over 50% and delayed the need for liver transplantation by 580 and 951 days, respectively. These are the longest reported survival times without LT in CNS1 patients treated with LCT. However, the loss of graft function was temporally associated with the emergence of donor-specific antibodies (DSAs), highlighting the need for further research to assess the impact of DSAs on hepatocyte engraftment and the long-term success of this approach.

As shown in Table 3, 14 CNS1 patients worldwide have undergone LCT, with 71% of them using this therapy as a bridge to liver transplantation [22,55,56,57,58,59,60,61,62,63]. LT was performed 4 to 31 months after the last LCT procedure [55]. Despite promising outcomes, LCT is still regarded primarily as a temporary measure to delay LT rather than a definitive treatment. The procedure offers a less invasive alternative that can significantly improve the quality of life and extend the period before liver transplantation is needed. However, further studies are required to enhance the long-term survival of transplanted hepatocytes and to evaluate the full potential of LCT in CNS1 treatment.

## 8. Gene Therapy

CNS is an ideal target for gene therapy because it is a monogenic disorder caused by mutations in the *UGT1A1* gene, leading to the inability to conjugate bilirubin. Since the liver is the primary site of *UGT1A1* expression and has a fenestrated endothelium, it allows easy access for gene transfer vectors through the bloodstream [1]. Over the last few decades, significant research has been conducted to identify the most effective vectors for applying gene therapy to CNS, primarily using the Gunn rat model [64]. The Gunn rat is a natural mutant that lacks UGT1A1 activity, making it an excellent model for studying CNS.

Recombinant adeno-associated virus (AAV) vectors have emerged as the most promising tools for gene replacement therapy, particularly in liver disorders like CNS. These vectors are favored because of their high safety profile and ability to deliver genetic material effectively [65]. AAV-mediated gene therapy is especially promising for adult patients, but it faces challenges in pediatric and juvenile patients because of the possibility of vector dilution in a growing liver, where hepatocyte proliferation can dilute the effects of the therapy [20]. Another challenge is the pre-existing immunity in some patients who have developed antibodies against the viral capsid, which can reduce the effectiveness of the therapy. As a result, researchers are actively exploring immunomodulatory strategies to overcome this barrier. In 2006, Seppen et al. [64] compared the efficiency of various AAV serotypes in the transduction of rat hepatocytes. Their study showed that AAV1 vectors were the most efficient in correcting *UGT1A1* deficiency in vivo. The level of *UGT1A1* correction achieved would be therapeutic in humans. However, the study also found large macroscopic lipid lesions of uncertain origin in all AAV-treated animals, which raised concerns about potential side effects. More recently, in 2018, Collaud et al. [66] conducted preclinical studies using an AAV8 vector carrying the human *UGT1A1* transgene (AAV8-hUGT1A1) under the control of a liver-specific promoter. The results demonstrated long-term safety and efficacy in correcting the *UGT1A1* deficiency in animal models, supporting the translation of this therapy to clinical trials. Toxicology and biodistribution studies confirmed that the AAV8-hUGT1A1 vector is well-tolerated and effective over time, paving the way for clinical application in humans.

Currently, two clinical trials are underway to evaluate gene transfer via recombinant viral vectors in CNS1 patients (clinicaltrials.gov: NCT03466463 and NCT03223194). These trials represent a significant step forward in the potential treatment of CNS1 using gene therapy. Interestingly, D’Antiga et al. recently published the results of the dose-escalation portion of the NCT03466463 study, which evaluated the safety and efficacy of a single intravenous infusion of AAV8-hUGT1A1 in patients with CNS who were being treated with phototherapy [67]. Five patients received a single infusion of the gene construct (GNT0003) as follows: two received 2 × 10^12^ vector genomes (vg) per kilogram of body weight and three received 5 × 10^12^ vg per kilogram. No serious adverse events were reported in patients treated with the gene therapy vector GNT0003 in the small study. Patients who received the higher dose had a decrease in bilirubin levels and did not receive phototherapy at least 78 weeks after vector administration [67]. Another potential approach to gene therapy is the transplantation of autologous hepatocytes that have been genetically corrected ex vivo. This strategy could serve as an alternative to full liver transplantation. Birraux et al. [68,69] proposed a novel ex vivo gene therapy technique called “SLIT” (Selective Liver Infusion Therapy). This process involves the isolation of hepatocytes from the patient, genetic correction via transduction with an HIV-1-derived lentiviral vector carrying the *hUGT1A1* transgene, and subsequent transplantation back into the patient. Lentiviral vectors are advantageous because they integrate into the genome of target cells and can transduce terminally differentiated cells, allowing for long-term therapeutic correction.

In preclinical studies, hepatocytes isolated from a CNS1 patient were successfully transduced with lentiviral vectors encoding the *hUGT1A1* gene. Following transplantation into immunodeficient mice, the corrected hepatocytes engrafted long-term (up to 26 weeks) and expressed the therapeutic protein. The study suggests that ex vivo gene therapy using lentiviral vectors could offer a permanent solution for CNS1 patients by providing a renewable source of corrected hepatocytes, potentially circumventing the need for liver transplantation.

In conclusion, gene therapy offers a promising approach to treating CNS, particularly for patients who cannot undergo liver transplantation or as an alternative to lifelong phototherapy. Advances in AAV-mediated gene delivery, immunomodulatory strategies, and ex vivo gene correction using lentiviral vectors are bringing gene therapy closer to clinical reality for CNS1. While challenges remain, including vector dilution in growing livers and pre-existing immunity, ongoing clinical trials and research efforts are laying the groundwork for a future where CNS1 can be effectively treated with gene therapy.

## 9. Conclusions

CNS consists of CNS1 and CNS2, both forms caused by mutations in the *UGT1A1* gene. CNS1 is the more severe form, characterized by a complete absence of *UGT1A1* activity, leading to extreme unconjugated hyperbilirubinemia and a high risk of irreversible brain damage (kernicterus). Treatments like phototherapy and plasmapheresis are only temporary, and LT remains the only definitive cure. The timing of LT is critical and must occur before neurological damage, though balancing early intervention with the risks of surgery and lifelong immunosuppression is challenging. Alternative transplantation methods such as living donor or domino transplantation offer potential solutions to donor shortages.

CNS2 is milder, with some residual enzyme activity, and patients typically respond well to phenobarbital and occasional phototherapy, with LT rarely needed. CNS2 patients usually have a better prognosis and quality of life compared with CNS1 patients.

Future directions include gene therapy, which holds promise for correcting the underlying genetic defect, particularly in CNS1, though challenges like vector dilution and immune responses must be overcome. Autologous hepatocyte transplantation offers another alternative, avoiding the complications of organ rejection. As the understanding of CNS continues to evolve, the goal is to offer CNS1 and CNS2 patients therapies that not only extend life but also improve its quality by minimizing the burden of lifelong treatments and preventing irreversible neurological damage.

## Figures and Tables

**Table 1 ijms-25-11006-t001:** Therapeutic strategies for Crigler–Najjar syndrome (CNS).

Treatment	Indication	Duration/Dose	Efficacy
Phototherapy	Soon after birth	12.4 ± 0.8 h	Temporary, most patients need daily cycles
Phenobarbital	First-line therapy in CNS2 patients	2 mg/kg/dose two–three times per day	Good response if taken lifelong
Orlistat	CNS1, CNS2	33–66 mg/m^2^ during meals for 4–6 weeks	Patients with lower dietary fat intake and low body mass index
Mesoporphyrin	CNS1, CNS2	Mesoporphyrin in a single dose of six micromoles per kilogram body weight, given intramuscularly	Lower plasma bilirubin level, lower frequency of severe hyperbilirubinemia, decreased need for phototherapy, shorter duration of hospitalization

**Table 2 ijms-25-11006-t002:** Therapeutic strategies in the main published studies.

Article	Study Population	Diagnosis	Treatment: First and Second Line	Outcome
Ihara et al. [34]	19-year-old Japanese man with CNS1	CNS1 diagnosed on liver biopsy (total absence of enzymatic activity) and at the age of 15 y sequencing *UGT* gene	-Phototherapy (>10 h every night) from a few days of life to 16 y. -Plasmapheresis performed with an anion-exchange resin (bilirubin-adsorbent column) from 16 y, for 6 months, two times/week (duration: 3 h).	Plasma perfusion could barely remove plasma the bilirubin synthesized each day. At 17 y, the patient received liver transplantation.
Yilmaz et al. [36]	15-year-old male	CNS2	-Phenobarbital (2 mg/kg/day) for 3 days p.o. produced 27% decrease in indirect bilirubin level. -Fenofibrate (250 mg/day os) for 1 month.	At the end of treatment with fenofibrate serum, indirect bilirubin was found to be unchanged.
Hafkamp et al. [37]	16 patients (pediatric > 7 years old and adult)	7 CNS1, 9 CNS2	Adults: 60–120 mg during meals; children: 33–66 mg/m^2^ during meals.	The decrease was clinically relevant (more than 10% decrease) in six patients (40%).
Shi et al. [32]	14-year-old female patient from Bangladesh	CNS2	Heterozygous for two different *UGT1A1* mutations: (1) Three nucleotide insertion in the HNF-1α binding site. (2) Two nucleotide deletion in exon 1 of *UGT1A1* gene (novel) results in a premature stop codon.	Not responsive to treatment with phenobarbital.
Strauss et al. [4]	20 patients diagnosed by sequencing the *UGTA1A* gene 19 patients had CNS1 phenotype (8 moths to 21 yo)	Seventeen patients (Amish or Mennonite descent) homozygous for a 222C→A mutation in exon 1 of *UGT1A1* that resulted in a stop codon (Y74X) and complete absence of transferase activity	Continuous high-intensity phototherapy. Four patients treated with orthotopic liver transplantation.	A systematic approach to neonatal screening, light dosing, and kernicterus prevention can assure that children and adults proceed to transplantation in good neurological health.
Liu et al. [7]	4-month-old Chinese boy	Compound heterozygous mutations: a missense mutation c.211G>A (p.G71R) in the first exon and a synonymous mutation c.1470C>T (p.D490D) in the fifth exon	Phototherapy (66 h) for 7 days and phenobarbital (5 mg/kg/day) for 3 days.	Followed until 4 months of age, no more treatment needed.
Gailite et al. [33]	17-year-old Caucasian male	Four different variants in the *UGT1A1* gene: g.3664A > C; g.4963_4964TA; g.5884G > T; g.11895_11898del.	Phenobarbital	The rs3755319 variant explains neonatal hyperbilirubinemia, the A(TA)7TAA variant explains hyperbilirubinemia, and two other variants are reported in a compound heterozygous state in patients with CNS2.
Suresh et al. [38]	84 cases 86 controls (from three single center studies)	Neonatal unconjugated hyperbilirubinemia	Mesporphyrin in a single dose of six micromoles per kilogram body weight, given intramuscularly.	Lower plasma bilirubin level, lower frequency of severe hyperbilirubinemia, decreased need for phototherapy, shorter duration of hospitalization.

Crigler–Najjar syndrome (CNS).

**Table 3 ijms-25-11006-t003:** Studies reporting liver cell transplantation (LCT).

Reference Study	Age/Weight at LCT	Hepatocytes Transplanted	Decrease in Bilirubin	Final Outcome
Fox et al. [55]	10 y/37 kg	7.5 × 10^9^ (0.2 × 10^9^/kg)	50%	LT after 4 years
Darwish et al. [57]	8 y/NA	7.5 × 10^9^ (NA)	40%	LT after 20 months
Allen et al. [60]	7 y/23 kg	1.4 × 10^9^ (0.06 × 10^9^/kg)	40%	LT after 11 months
Ambrosino et al. [42]	9 y/30 kg	7.5 × 10^9^ (0.25 × 10^9^/kg)	50%	LT after 5 months
Dhawan et al. [22]	18 m/NA	4.3 × 10^9^ (NA)	50%	LT after 8 months
	3 y/7 kg	2.1 × 10^9^ (0.29 × 10^9^/kg)	30%	LT not performed
Stéphenne et al. [61]	9 m/NA	NA (033 × 10^9^/kg)	NA	NA
Khan et al. [62]	2 y/NA	0.015 × 10^9^ (NA)	NA	NA
Quaglia et al. [56]	1 y/NA	4.34 × 10^9^ (NA)	NA	LT after 6 months
Lysy et al. [63]	9 y/38 kg	6 × 10^9^ (0.16 × 10^9^/kg)	35%	LT after 6 months
	1 y/7.4 kg	2.6 × 10^9^ (0.35 × 10^9^/kg)	65%	LT after 4 months
Ribes-Koninckx et al. [58]	7 m/NA	6.7 × 10^9^ (NA)	NA	LT not performed
Jorns et al. [59]	13 y	11.2 × 10^9^	50%	LT after 19 months
	11 y	5.3 × 10^9^	50%	LT after 31 months

Liver transplant (LT); not available (NA).

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
