# Peer review of "Therapeutic Options for Crigler–Najjar Syndrome: A Scoping Review"

_ijms, 2024, doi:10.3390/ijms252011006_

Round 1

Reviewer 1 Report

Comments and Suggestions for Authors

Here are may comments:

the search methodology should be moved to the methods section. It does not belong to the introduction. 

Please indicate the authors who carried out the literature search. 

The sentence "However, it is more prevalent in populations with a high frequency of UGT1A1 mutations due to a founder effect, such as the Old Order Amish and Mennonite communities" is unclear since the disease is genetically segregated. I'd say that the disease is more prevalent in genetically segregated populations or populations with high inbreeding, like .....

Gilbert syndrome is a disease spectrum. Some forms are related to transmembrane ligands altered activity, while others are closely correlated to Crigler Najar. Please rephrase.

It's necessary to add that unconjugated bilirubin is transported with albumin. This is needed better to capture the clinical meaning of the B/A ratio. 

Gut bacteria convert conjugated bilirubin into urobilinogen. This is transported into the blood (entero-hepatic circulation) and excreted in the urine. Urobilin is the oxidized form of urobilinogen.

Likewise, urobilinogen is converted by the colonic flora into stercobilinogen. Stercobilin is the oxidized form in feces.

 Donor availability is not a risk for liver transplantation with less favorable results. I'd suggest the authors change their sentence to state that the number of pediatric donors limits the availability of liver transplantation. However, surgical strategies like liver splitting can offer great advantages to pediatric recipients.

There is no difference between ALT and APOLT in describing the surgical strategies of liver transplantation. I'd suggest the authors state that liver transplantation may be from a deceased donor using a whole-size or split graft or a living donor, the latter usually from related donors (parents). It's simpler and more intuitive. 

LT is recommended after the first year of life because its results improve. Recipients below 1 year fare worse in general. 

In Table 3, Quaglia is misspelled.

Comments on the Quality of English Language

Some minor flaws require consideration. 

Author Response

Re: Thank you for your evaluation. We revised the manuscript according to your comments.

Here are may comments:
the search methodology should be moved to the methods section. It does not belong to the introduction. 
Re: Done as suggested (p. 2).

Please indicate the authors who carried out the literature search. 
Re: Indicated (p. 2).

The sentence "However, it is more prevalent in populations with a high frequency of UGT1A1 mutations due to a founder effect, such as the Old Order Amish and Mennonite communities" is unclear since the disease is genetically segregated. I'd say that the disease is more prevalent in genetically segregated populations or populations with high inbreeding, like .....
Re: Revised as suggested (p. 2).

Gilbert syndrome is a disease spectrum. Some forms are related to transmembrane ligands altered activity, while others are closely correlated to Crigler Najar. Please rephrase.
Re: Rephrased according to your comment (p. 2).
It's necessary to add that unconjugated bilirubin is transported with albumin. This is needed better to capture the clinical meaning of the B/A ratio. Gut bacteria convert conjugated bilirubin into urobilinogen. This is transported into the blood (entero-hepatic circulation) and excreted in the urine. Urobilin is the oxidized form of urobilinogen. Likewise, urobilinogen is converted by the colonic flora into stercobilinogen. Stercobilin is the oxidized form in feces.
Re: The paragraph has been revised according to your comments (p. 3).

Donor availability is not a risk for liver transplantation with less favorable results. I'd suggest the authors change their sentence to state that the number of pediatric donors limits the availability of liver transplantation. However, surgical strategies like liver splitting can offer great advantages to pediatric recipients.
Re: Clarified as suggested (p. 11).

There is no difference between ALT and APOLT in describing the surgical strategies of liver transplantation. I'd suggest the authors state that liver transplantation may be from a deceased donor using a whole-size or split graft or a living donor, the latter usually from related donors (parents). It's simpler and more intuitive. 
Re: Revised as recommended (pp. 10-11).

LT is recommended after the first year of life because its results improve. Recipients below 1 year fare worse in general. 
Re: Clarified as suggested (p. 11).

In Table 3, Quaglia is misspelled.
Re: Corrected (p. 14).

Reviewer 2 Report

Comments and Suggestions for Authors

The paper is precisely written, exploring various issues related to Crigler-Najjar syndrome. It is based on many up-to-date publications. What I miss here is a short explanation concerning the purpose of the article in the Abstract and a similar comment (maybe a little bit longer) in the Introduction, emphasizing the meaning of this pathology, the need of looking for new therapies - to describe precisely, why the authors decided to write such a manuscript. Additionally, I think that the inclusion of additional figure presenting a general scheme of Crigler-Najjar syndrome would be also interesting for readers.

Author Response

The paper is precisely written, exploring various issues related to Crigler-Najjar syndrome. It is based on many up-to-date publications. What I miss here is a short explanation concerning the purpose of the article in the Abstract and a similar comment (maybe a little bit longer) in the Introduction, emphasizing the meaning of this pathology, the need of looking for new therapies - to describe precisely, why the authors decided to write such a manuscript. Additionally, I think that the inclusion of additional figure presenting a general scheme of Crigler-Najjar syndrome would be also interesting for readers.

Re: Thank you very much for your positive evaluation. We added further explanation on the purpose of this manuscript in the Abstract (p. 1) and the Introduction (pp. 1-2). We did not add a Figure because we think that the three Tables summarize in an appropriate way the main available studies. Moreover, we further improved the text according to the Edotor’s comments and the other reviewer’s suggestions.